# Mindfulness Practice Reduces Hair Cortisol, Anxiety and Perceived Stress in University Workers: Randomized Clinical Trial

**DOI:** 10.3390/healthcare11212875

**Published:** 2023-10-31

**Authors:** Edilaine Cristina da Silva Gherardi-Donato, Larissa Bessani Hidalgo Gimenez, Maria Neyrian de Fátima Fernandes, Riccardo Lacchini, Elton Brás Camargo Júnior, Kranya Victoria Díaz-Serrano, Melissa Melchior, Raquel García Pérez, Jorge Riquelme-Galindo, Emilene Reisdorfer

**Affiliations:** 1Ribeirão Preto College of Nursing, University of São Paulo, Avenida Bandeirantes, 3900, Vila Monte Alegre, Ribeirão Preto 14040-902, SP, Brazil; larissabhgimenez@gmail.com (L.B.H.G.); rlacchini@eerp.usp.br (R.L.); 2Nursing Department, Federal University of Maranhão, Avenida Principal, 100 Residencial Dom Afonso Felippe, Imperatriz 65915-240, MA, Brazil; neyrian.maria@ufma.br; 3Faculty of Nursing, University of Rio Verde, Fazenda Fontes do Saber, Rio Verde 75901-970, GO, Brazil; eltonbrasjr@unirv.edu.br; 4School of Dentistry of Ribeirão Preto, University of São Paulo, Avenida do Café, s/n, Vila Monte Alegre, Ribeirão Preto 14040-904, SP, Brazil; dkranya@forp.usp.br (K.V.D.-S.); memelc07@gmail.com (M.M.); 5Faculty of Health Sciences, University of Granada, Avenida de la Ilustración, 18016 Granada, Spain; raquelgp@ugr.es; 6HLA Vistahermosa Hospital, Avenida Denia, 03015 Alicante, Spain; jrgrqlm@gmail.com; 7Department of Professional Nursing and Allied Health, Faculty of Nursing, MacEwan University, 104 Avenue NW 10700, Edmonton, AB T5J 4S2, Canada; reisdorfere@macewan.ca

**Keywords:** mindfulness, hair cortisol, stress, workers, mental health

## Abstract

Background: Anxiety and stress are common mental health conditions reported by university workers. Practices of mindfulness represent one promising approach as an effective and feasible means to reduce stress, improve mental health and promote well-being; however, there are no clinical trials that have combined long-term stress biomarkers (hair cortisol) and psychometric assessments in a sample of university workers. Objective: This study investigated the effectiveness of a mindfulness-based program on long-term stress, by measuring hair cortisol concentration and perceived stress and anxiety among workers who were undergoing high levels of stress. Method: We conducted a randomized clinical trial at work among the employees of a public university. We compared a group that received the eight-week mindfulness intervention with the wait list group who received no intervention. Results: A total of 30 participants were included in the study, with n = 15 subjects in the intervention group and n = 15 in the control group. Hair cortisol, perceived stress and anxiety significantly reduced after the intervention compared to the control group, which had no appreciable decline in the measured variables. Conclusion: This clinical trial showed the effectiveness of a mindfulness program on mental health psychometric measures (perceived stress and anxiety) and on a long-term stress biomarker (hair cortisol). It can be concluded that an eight-week mindfulness program could be implemented as an effective strategy to reduce stress biomarkers (hair cortisol) as well as perceived stress and anxiety, improving the mental health of university workers.

## 1. Introduction

University workers constitute a population with significant rates of work overload, job dissatisfaction, occupational stress [1,2], anxiety and the use of non-assertive coping strategies such as alcohol abuse [3]. Perceived stress is associated with higher levels of anxiety and poor quality of life [4]. Strategies for managing and reducing stress can also decrease the occurrence of clinically significant anxiety [5].

There is substantial scientific evidence that mindfulness practices reduce subjective stress and increase the sense of well-being in workers from different sectors, providing particular support for people suffering from stress [6,7,8,9]. Mindfulness has been associated with lower perceived stress and higher work engagement in both cross-sectional and longitudinal analyses [10]. 

Also, the practice of mindfulness has been reported as a strategy to reduce levels of anxiety and improve quality of life. The effects of the practice in reducing anxiety symptoms have been explained by the impact of a mindful mental state on reappraisal, worry and rumination [11]. Given the main mental health problems related to stress among university workers, a study that targets this population using mindfulness programs seems highly relevant. 

Persistent or prolonged stress can have deleterious effects on bodily mechanisms, and there are many systems in the body that individually or collectively regulate the level of stress, including the hypothalamic-pituitary-adrenal axis (HPA axis), autonomic nervous system (ANS) and immune system. The HPA axis responds to stressors by secreting cortisol, a hormone associated with psychological stress, making the HPA axis a system that responds to psychosocial stress and interacts with the ANS and immune system. Cortisol is physiologically measurable through analysis of saliva, blood, urine and hair samples [12].

Several studies have used cortisol, considered a biomarker of physiological stress, as a measure of the effect of mindfulness-based interventions. The accurate measurement of cortisol in saliva has been reported as a methodological obstacle due to the variability resulting from the circadian rhythm (i.e., diurnal fluctuation) and the influence of oral conditions, among other factors [13]. However, hair sampling is a method considered less susceptible to the methodological obstacles reported in the assessment of cortisol, and it is more accurate for studies that aim to verify the effects on physiological factors relevant to the neuroendocrine mechanism that mediates the effects of stressors by regulating numerous physiological processes, such as the hypothalamic–pituitary–adrenal axis (HPA axis) [14]. Hair cortisol can reflect prolonged stress exposure, making it an ideal measure for assessing vulnerable populations such as university workers.

Assessment of perceived stress is considered equally important, as the stress response is highly influenced by the person’s perception of the stressor. The Perceived Stress Scale (PSS) is one of the most common measures for assessing global stress perceptions. It measures the degree to which an individual perceives his/her life as uncontrollable, unpredictable and overloaded within the past month [15]. The PSS is closely linked with measures of psychological stress and self-reported health (depressive and physical symptomatology) [16]; it is also correlated with biological markers of stress and disease [17]. The Perceived Stress Scale (4-, 10-, and 14-item versions), with its psychometric properties, has been translated into different languages, including Brazilian Portuguese [18].

A recent clinical trial that evaluated the effect of mindfulness-based interventions for workers presented results for the measure of perceived stress in groups that underwent short- and long-term mindfulness programs, considering the potential dose–response effects of mindfulness training at work [19]. In addition, several studies conducted with primary and secondary teachers also pointed to the effects of mindfulness as an effective stress management strategy [20,21,22].

However, no studies were found that combined biological measurement and psychometric assessment in a sample of university workers. Therefore, it is necessary to undertake studies providing evidence of the effectiveness of the mindfulness program, in order to support the recommendation of the practice of mindfulness for the care of university workers who present a high level of stress and associated anxiety.

This study investigated the effect of a mindfulness-based program on hair cortisol, perceived stress, mindfulness levels, and anxiety among workers who were undergoing high levels of stress.

## 2. Materials and Methods

### 2.1. Trial Design 

A randomized clinical trial (RCT) was conducted in accordance with the recommendations of the Consolidated Standards of Reporting Trials (CONSORT) [23]. It was an intervention study of a controlled, randomized, parallel, 2-arm clinical trial type (intervention group with intervention versus wait list control group without treatment). The study was registered in the Brazilian Clinical Trials Registry (ReBec), Trial Registration Number (TRN): U1111-1179-7619. The procedures for data collection and analysis were completed in May 2019.

### 2.2. Participants and Recruitment

Following approval by the Brazilian Ethics Committee (protocol number 58376016.00000.5393) in accordance with The Code of Ethics of the World Medical Association (Declaration of Helsinki), technical-administrative workers from a University in the State of São Paulo, Brazil were recruited to take part in the study (1704 workers). All workers expressed their willingness to participate and signed an informed consent form. A team of researchers, during the period from March to December 2017, personally approached all participants in their work environment and applied a sociodemographic questionnaire and the Perceived Stress Scale (PSS 14) [18]. To ensure data accuracy, we excluded certain workers from the research. Those who were not found at their workplace after three attempts by the research team (n = 163) or who were on vacation (n = 192) or sick leave (n = 114) during the data collection period were excluded. Incomplete materials (n = 42) and subjects who did not respond to the questionnaires (n = 264) were excluded from the study, composing the losses of the screening process. 

A total of 929 volunteers returned completed data collection. This accounted for 54.51% of the total non-teaching staff population on campus. The selected 929 participants answered a sociodemographic questionnaire and the Perceived Stress Scale (PSS 14) [18]. During the period from March to December 2017, the participants were personally approached in their respective workplaces by a team of researchers and invited to participate. Workers were eligible for the clinical trial if they met the criteria for high perceived stress (score equal to or greater than 23 points), as measured by PSS 14 [18]. All workers were invited to perform the stress assessment. Workers who met one or two of the following exclusion criteria were not eligible for enrollment in the study: regular meditation practice (at least once a week in the past 12 months) and engagement in psychotherapy or mental health counseling. These exclusion criteria were used to reduce any potential confounding biases that could affect the intervention’s expected effect.

A total of 281 subjects met the inclusion and exclusion criteria for the study and were randomized to the intervention group (IG) (participated in the 8-week mindfulness program) or the control group (CG) (placed on a waiting list). Following the 8-week assessment period, the workers from the control group were given the option of participating in the 8-week mindfulness program. 

### 2.3. Randomization and Blinding

Sample size calculations were performed by a statistician and indicated that 76 individuals were needed in the intervention (38) and control (38) groups.

The sample of the study was determined utilizing a convenience method based on participants’ availability. The number of enrolled participants was defined considering their available time to commit to the eight weeks of intervention and data collection, as well as the cost-effectiveness related to human resources, infrastructure and laboratory tests necessary to respond to the objective of the study. For the pairing of the groups, age, sex and level of work (basic, technical and higher) were considered, and participants were randomized in a 1:1 ratio. Considering all the requirements mentioned above, available participants (n = 42) were randomly assigned to either an intervention group (IG = 22) to receive the 8-week mindfulness program or to a control group (CG = 20), which received no intervention and remained on a waiting list.

The allocation list was generated by a professional statistician using a random number table, and the sequence was concealed until the participants were assigned. Recruitment was conducted by the researcher, and instructors who conducted the intervention were blinded to group assignment. Participants were randomly allocated to the intervention group (IG) and control group (CG), without the option to choose their group. Consequently, the selected participants were invited to participate in the second phase of the study and provided with a new informed consent form.

Data regarding the variables investigated (hair cortisol, perceived stress, symptoms of anxiety and level of mindfulness) were collected 2 weeks before the beginning of the study and again after the end of the study (12 weeks after the first data collection).

This intervention could not be masked (participants knew which group they were in and instructors knew they were performing the intervention); thus, to avoid research bias, the instructors who performed the intervention had no contact with the data or any approach of the research-related procedures. The statistical professional who performed randomization and analysis was blinded to the intervention.

### 2.4. Psychometric Assessment

Perceived stress was assessed by the Perceived Stress Scale 14-item version (PSS 14). The PSS measures the degree to which individuals perceive situations as stressful [18]. The total score on the scale can range from 0 (no stress) to 56 (extreme stress) [15,18].

Anxiety was assessed by the Beck Anxiety Inventory (BAI), created by Beck et al. (1988) [24]. It consists of 21 questions about how the individual has been feeling in the last week, expressed in common anxiety symptoms (such as sweating and feelings of distress). Each question has 4 possible answers; the one that most closely resembles the individual’s mental state should be marked and ranges from 0 to 63.

Mindfulness was assessed by the Five Facet Mindfulness Questionnaire (FFMQ-BR). This instrument measures levels of mindfulness in a multidimensional way [25]. Although listed as five facets, the authors who validated the Brazilian version recommend that the instrument should be analyzed considering seven facets, namely (1) non-judging of inner experience, (2) acting with awareness–autopilot, (3) observing, (4) describing–positive formulation, (5) describing–negative formulation, (6) non-reactivity to inner experience, and (7) acting with awareness–distraction [25].

### 2.5. Hair Sample for Cortisol Measurement

Hair was sampled from the vertex posterior of workers’ heads. Using sterile scissors, a strip of approximately 1.5 inches (3 cm) of hair was cut at the scalp where there was a hair length of at least 1.5 inches (3 cm). This 3 cm length is considered to represent a 3-month retrospective measure of hair cortisol, assuming a 1 cm/month growth rate of hair; it is a commonly used length for sampling [26,27]. Samples were stored in plastic vials labeled only by ID in a locked room and at room temperature according to the storage standards [28] for laboratory assays.

### 2.6. Hair Processing and Cortisol Assay

The samples were sent for assay to the Laboratory Specialized in Scientific Analysis (LEAC) [26,27]. Starting with ∼50 mg of hair, two washes with 40 mL of water followed by two washes with 40 mL of isopropanol on a (orbital shaker) plate rotating at 130 rpm for 3 min per wash were performed.

After each set of washes, the hair samples were cut into small pieces using small surgical scissors. Samples were put into disposable glass scintillation vials, and HPLC-grade methanol was added at a volume of 100 μL/mg of hair. Next, samples were sonicated for 30 min followed by a 24 h incubation period at 50 °C. After incubation, samples were centrifuged for 30 min at 3000 rpm, then the supernatant was transferred into separate glass tubes and the methanol was evaporated under a gentle stream of nitrogen. For samples where less than 30 mg of hair was available, smaller volumes of the extract were aliquoted, corresponding to 5, 10 or 15 mg of hair. Once the methanol was removed, the dry pellet was resuspended in 150–250 μL of phosphate buffered saline (PBS) at pH 7.0. Samples were vortexed for 1 min, followed by another 30 s until they were well mixed. For cortisol measurement in the extracts, we used a commercially available salivary cortisol enzyme-linked immunosorbent assay (ELISA) (Cat. #KAPDB290 (CTS)—Lot 150810—DiaSource) according to the manufacturer’s instructions. The intra- and inter-assay coefficient of variance was below 10.3%.

### 2.7. Intervention: 8-Week Mindfulness Program 

Participants randomized to the 8-week mindfulness program attended weekly in-person 120 min mindfulness-based practice group sessions for 8 consecutive weeks. The instructor progressively promoted and guided formal practices with the participants and mediated the exchange of experiences in each weekly session. Such formal practices require cognitive efforts, especially attention, which gradually evolve throughout the eight weeks, based on different personal processes focused on breathing, the body, sensations, sounds, thoughts and emotions. In addition, there was a 4 h immersion session in a larger space with an added natural area (trees, plants and grass floor). The program was led by health professionals, skilled facilitators and mindfulness instructors who were certified experts in providing guidance for learning mindfulness practices. Their expertise increased the chances of successful attainment of a state of mindfulness by the participants [29]. The program consisted of mindfulness practice training in a secular context, following the foundations of the Mindfulness-Based Stress Reduction (MBSR) program [30] and recommendations from the UK Network for Mindfulness-Based Teachers Good Practice Guidelines for Teaching Mindfulness-based Courses [31]. The following topics were discussed during the 8 sessions: change is easy, breathing is natural, keeping the body conscious, just be, moment by moment, welcoming emotions, living mindfulness, cultivating peace.

During face-to-face meetings, participants were instructed to perform the practices daily at home for the rest of each week. Relevant readings and audio material were offered to support the practice. Audio was made available to participants in diverse ways: e-mail, WhatsApp^®^ or Compact Disc (CD), according to the preference indicated by the participant. Audio files contained a guided practice (5-, 10-, 20-, and 30-min body scans and guided meditation with different positions and anchors) recorded by the mindfulness instructors. A simple habit change was guided each week with the purpose of enhancing the recognition of automatic behaviors. A diary was also offered weekly, in which participants were encouraged to register their practices at home. The diary was optional and not used in the present analysis. The main purpose was to stimulate and organize the integration of practices in the participants’ daily lives.

### 2.8. Statistical Approach

To compare hair cortisol, perceived stress (PSS14), anxiety (BAI) and mindfulness (FFMQ) at pre-intervention (T0) and post-intervention (T1) times in each group, the Wilcoxon test for related samples was used. To compare the differences between the intervention group and control group, the non-parametric Wilcoxon rank sum test was used. The data were analyzed using the statistical software STATA 11 edition.

## 3. Results

A total of 30 volunteers completed the study, with a loss of 28.6% of participants who had started the study (42): seven from the intervention group and five from the wait list control group. Follow-up was lost due to withdrawal (four participants reported a lack of time and one participant moved to another city), incomplete intervention (three participants completed less than 60% of the intervention program), and loss of the post-intervention assessment period (four participants were unavailable for post-intervention data collection). The demographic data and hair cortisol, perceived stress, anxiety and mindfulness baseline scores are presented by group in Table 1.

No differences were found in baseline levels of hair cortisol or any of the psychological outcome variables (mindfulness, perceived stress and anxiety), indicating that the randomization was successful. There was no significant difference in baseline cortisol level for age, education, working hours and working years (all *p* > 0.05) (Table 1).

Participants who completed the eight-week mindfulness program had their hair cortisol levels significantly reduced. The Wilcoxon signed rank test results show that the hair cortisol median presented by the intervention group after the intervention was reduced by 3.9 pg/mg (*z*: 2.953, *p*: 0.003), while the hair cortisol of the control group showed no significant change during the studied period. Also, the psychometric evaluation of the participants who completed the mindfulness program showed a significant decrease in the median perceived stress (*z*: 2.160, *p*: 0.031) and symptoms of anxiety (*z*: 2.074, *p*: 0.038) as well as increased levels of mindfulness (*z*: 1.992, *p*: 0.046) post-intervention (Table 2).

The Wilcoxon rank sum test presents a comparison between the control and intervention groups at the median level of cortisol, anxiety, perceived stress and level of mindfulness before and after the intervention (Table 2). The results show that the intervention decreased the median hair cortisol values (*z*: −1.805, *p*: 0.071) as well as symptoms of perceived stress (*z*: −2.599, *p* < 0.001) and median anxiety (*z*: −2.082, *p*: 0.037) when compared with the control group. The intervention group also experienced an increase in the level of mindfulness (*z*: −1.848, *p*: 0.065). This result did not occur in the control group, and no significant difference was found between the medians at baseline and after eight weeks in that group.

The result regarding the effectiveness of the mindfulness program in relation to the control group, considering the evolution of the study effect variables, revealed that the intervention reduced the risk of worsening hair cortisol by 88.8%, perceived stress by 54.6% and anxiety by 50.0%. The percentage of worsening symptoms was 6.7% in the intervention group versus 60.0% in the control group for hair cortisol, 33.3% versus 77.3% for perceived stress, and 23% versus 53.3% for anxiety symptoms (Table 3).

## 4. Discussion

As a context that generates stressful situations, the university environment has drawn the attention of the scientific community, which has sought ways to better understand the diverse factors that influence this condition and to promote strategies that the university community can employ to improve stress management, such as mindfulness interventions.

Since the literature points to the high levels of stress among university workers, and in order to make the evidence more robust, our study used instruments based on the individuals’ perceptions and integrated objective ways of measuring stress levels, which distinguishes our work from some earlier studies.

In this study, we evaluated the effects of an eight-week mindfulness program on stress and anxiety compared to the results of the wait-listed control group that did not receive the intervention. University workers with a high level of perceived stress were evaluated in both the intervention and control groups in relation to perceived stress, anxiety and hair cortisol levels, the latter being considered the most reliable biomarker of chronic stress and the most widely used validation test [12].

There was a significant effect of study conditions on changes in hair cortisol, perceived stress, and symptoms of anxiety. Workers had significant reductions in cortisol levels after the intervention compared with those in the control group, who had no appreciable decline. Perceived stress and symptoms of anxiety also reduced significantly, which did not occur in the control group.

The increase in the scores for the mindfulness assessment, which occurred only in the intervention group, shows that the intervention was successful, allowing this effect to be linked to the set of variables analyzed.

Our findings are consistent with previous studies that have evaluated the effect of a mindfulness program on perceived stress reduction in other settings and populations [6,7,8,9,21]. Mindfulness-based interventions promote brain responses at the level of regulation and reactivity. Thus, one explanation for understanding the reduction in stress levels resulting from the practice of mindfulness is that it promotes the regulation of reactivity to stress in the HPA axis [32].

Regarding anxiety symptoms, our results are also in line with previous evidence, which concluded that mindfulness-based programs are effective in reducing anxiety [33].

One of the original contributions of this study was to evaluate psychometric parameters (perceived stress and anxiety) together with a biological stress marker (capillary cortisol); thus, regarding the results in the evaluation of hair cortisol, the decrease in this long-term stress marker was also reported in a study carried out with undergraduate healthcare students [34]. Hair cortisol assessment is an excellent biomarker of chronic stress, correlating with salivary cortisol levels, which has been corroborated by several studies on stress in different populations [35]. The cortisol available in the hair represents the percentage of this biomarker that diffuses from the blood into the growing hair follicles and is incorporated into the hair without undergoing degradation [36,37].

One crucial discovery of this study is that the control group witnessed a substantial elevation in cortisol levels over the course of the study. This noteworthy finding emphasizes the significance of including all participants in the study, ensuring that everyone can reap the benefits of the intervention. The observed increase in cortisol levels within the control group underscores the potential impact of the intervention in mitigating stress and its related physiological responses.

In assessing the effectiveness of the treatment by the relative reduction rate in the risk of worsening symptoms, we highlight the percentage of participants in the control group who had increased hair cortisol (60.0%), compared with only one participant (6.7%) in the intervention group. 

Considering that the effectiveness of the treatment for perceived stress and anxiety was lower than that found for capillary cortisol, despite being relevant and positive, it is observed that the physiological result precedes the psychometric one. Since the BAI and PSS offer acute measures of psychological distress while hair cortisol measures more chronic stress, these might actually be reversed. This could be explained by the literature suggesting that mindfulness can actually increase one’s awareness of subjective anxiety and stress more acutely, since they are no longer avoiding it; this makes one better able to cope with it long-term but might mask some of the findings of self-reported data for these constructs [38]. It is valid to also consider the subjectivity inherent to the psychometric assessment using self-report scales [39] and, on the other hand, the unconscious maintenance of a condition that guarantees the participant’s eligibility for professional assistance on an intrapsychic level for these workers.

We offer evidence based on validated psychometric instruments and a reliable biomarker of chronic stress—the most widely used validation test, hair cortisol [12]—regarding the benefits of the mindfulness program, which contributes to the promotion of mental health and the prevention of illness among university workers. The implementation of mindfulness practices in the routine of these workers could constitute an important strategy for managing stress and its long-term consequences for the mental health and well-being of workers in the university context.

## 5. Conclusions

The mindfulness-based intervention for university workers sample showed improvement in a long-term stress biomarker, accessed by hair cortisol concentration, as well as in perceived stress and anxiety symptoms, measured by self-reported scales. The effectiveness of the intervention compared to the control group was highlighted and corroborated by the worsening of the condition in workers who did not receive support for stress management. Such results raise important questions about stress in university workers and its relationship with the conditions of the academic environment.

The eight-week mindfulness-based intervention program was relevant to the promotion of mental health and could contribute to the prevention of diseases in university workers. Further studies are needed to evaluate the effectiveness of the intervention program, including other mental health parameters.

Some limitations to consider are that the control group participants knew they were included in the wait list condition, and relevant data related to race and ethnicity were not collected. The small sample size limited the generalizability of the results. We considered that the sample size was limited by an intervention that lasted eight weeks, since that required a commitment not everyone was willing to make. Another possible limitation of the study is the dropout rate of 28.6%. Although traditional randomized controlled trials accept a dropout rate of up to 20% [40], it has been noted that in mindfulness studies, a rate of 16–29% is frequently reported [41].

## Figures and Tables

**Table 1 healthcare-11-02875-t001:** Demographic characteristics and levels of cortisol, perceived stress, depression and anxiety of stressed university workers (n = 30).

Variable	Intervention Group (n = 15)	Wait List Control Group (n = 15)	
Age, mean (SD)	41.4 (7.6)	40.5 (8.2)	*p* = 0.865
Education, number (%)			*p* = 0.406
High school	4 (26.7)	4 (26.7)	
Graduate	5 (33.3)	6 (40.0)	
Postgraduate	6 (40.0)	5 (33.3)	
Working hours, mean (SD)	39.7 (1.0)	39.3 (2.6)	*p* = 0.281
Working years, mean (SD)	14.8 (7.5)	12.3 (7.0)	*p* = 0.639
Level of mindfulness (FFMQ Score)	111.1 (17.7)	115.5 (17.7)	*p* = 0.367

FFMQ-BR: Five Facet Mindfulness Questionnaire (Brazilian Version); SD: standard deviation.

**Table 2 healthcare-11-02875-t002:** Comparison of the level of hair cortisol, anxiety and perceived stress in a mindfulness training intervention clinical trial before and after the intervention (n = 30).

Variable	CG (n = 15)	Wilcoxon Signed Rank Test (*p*)	EG (n = 15)	Wilcoxon Signed Rank Test (*p*)	Wilcoxon Rank Sum Test
**Hair cortisol**					
T0 median (SD)	15.6 (12.1)	*z*: −0.795, *p*: 0.426	18.9 (8.7)	*z*: 2.953, *p*: 0.003	*z*: 0.892, *p*: 0.372
T1 median (SD)	18.1 (13.0)		15.0 (5.3)		*z*: −1.805, *p*: 0.003
**Perceived stress**					
T0 median (SD)	31 (6.2)	*z*: 0.258, *p*: 0796	32 (8.2)	*z*: 2.160, *p*: 0.031	*z*: 0.021, *p*: 0.983
T1 median (SD)	32 (4.8)		24 (7.8)		*z*: −2.599, *p* < 0.001
**Anxiety**					
T0 median (SD)	11 (8.4)	*z*: 1.082, *p*: 0.279	10 (6.5)	*z*: 2.074, *p*: 0.038	*z*: −0.520, *p*: 0.603
T1 median (SD)	11 (6.8)		5 (5.3)		*z*: −2.082, *p*: 0.037
**Level of mindfulness**					
T0 median (SD)	119 (17.5)	*z*: −0.934 *p*: 0.3502	113 (15.6)	*z*: 1.992 *p*: 0.046	*z*: 0.512, *p*: 0.609
T1 median (SD)	113 (17.7)		127 (15.9)		*z*: −1.848, *p*: 0.065

Alpha criteria = 0.05; CG: Control Group; IG: Intervention Group; SD: standard deviation.

**Table 3 healthcare-11-02875-t003:** Treatment effectiveness of mindfulness program in intervention and control groups.

Variable	Intervention Group (n = 15)	Control Group (n = 15)	Treatment Effectiveness (RRR)
Increased hair cortisol			
Yes	1 (6.7%)	9 (60.0%)	88.8%
No	14 (93.3%)	6 (40.0%)	
Perceived stress			
Yes	5 (33.3%)	11 (77.3%)	54.6%
No	10 (66.7%)	4 (26.7%)	
Anxiety			
Yes	4 (26.7%)	8 (53.3%)	50.0%
No	11 (77.3%)	7 (43.7%)	

RRR = Relative reduction in the risk of worsening symptoms.

## Data Availability

The data presented in this study are available on request from the corresponding author. The data are not publicly available due to local ethical legislation.

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
