# Peer review of "Mindfulness Practice Reduces Hair Cortisol, Anxiety and Perceived Stress in University Workers: Randomized Clinical Trial"

_healthcare, 2023, doi:10.3390/healthcare11212875_

Round 1
Reviewer 1 Report
Comments and Suggestions for Authors
Thanks for letting me review this manuscript. It is nice to read about this intervention. However, there are several problems with the manuscript that raise significant questions about its readiness for publication at this time.
1. The structure of the introduction section is relatively loose, and the logic should be more rigorous.( P1-2 )
2. The structure of the introduction section is loose, and the logic should be more rigorous. The authors should clearly illustrate the research hypothesis and purpose in the introduction's last paragraph.( P 2 )
3. The author should describe the specific methods and parameters for calculating sample size in section of “2.3. Randomization and Blinding”. (P 3)
4. The authors should complement the name or topic of each session of the 8-Week Mindfulness Program.( P 5)
5. The authors only briefly summarized the statistical analysis results and needed to provide a detailed explanation in the section of discussion. ( P 7-9 )
Author Response
Dear Reviewer
We appreciate all you comments to improve our paper. Please find the responses to your suggestions attached.
Kind regards,
Authors

Reviewer 2 Report
Comments and Suggestions for Authors
Overall, this manuscript describes a study that aims to contribute both to the mindfulness literature as it pertains to stress and anxiety, and it contributes to the stress literature as it pertains to measuring physiological and self-reported stress. Given some clarification to methodology and descriptions of findings in certain areas, changes to wording in various places, and some references to some other literature in the Introduction and Discussion, the manuscript would be able to make a significant contribution to both.
Introduction
- There are a few other mindfulness studies for those in education environments that could be useful to reference that also measured biological and perceived measures of stress. Although these studies were done in primary and secondary schools as opposed to a university, they will be useful to put in the Introduction to help justify why it is important to measure both. These citations are below:
Flook, L., Goldberg, S. B., Pinger, L., Bonus, K., & Davidson, R. J. (2013). Mindfulness for teachers: A pilot study to assess effects on stress, burnout, and teaching efficacy. Mind, Brain, and Education, 7(3), 182-195.
Harris, A. R., Jennings, P. A., Katz, D. A., Abenavoli, R. M., & Greenberg, M. T. (2016). Promoting stress management and wellbeing in educators: Feasibility and efficacy of a school-based yoga and mindfulness intervention. Mindfulness, 7, 143-154.
Roeser, R. W., Schonert-Reichl, K. A., Jha, A., Cullen, M., Wallace, L., Wilensky, R., ... & Harrison, J. (2013). Mindfulness training and reductions in teacher stress and burnout: Results from two randomized, waitlist-control field trials. Journal of educational psychology, 105(3), 787.
Taylor, S. G., Roberts, A. M., & Zarrett, N. (2021). A brief mindfulness-based intervention (bMBI) to reduce teacher stress and burnout. Teaching and Teacher Education, 100, 103284.)
Methods
- The description of the inclusion into the intervention part of the study is confusing since the inclusion/exclusion criteria listed (regular meditation practice and/or receiving psychological treatment) was already described for the first phase (or at least it seems that way). It would be good to include the reason for that (to reduce any confounding biases) to the part where it is first mentioned, and then it could be removed when discussing the second study.
- In line 138, it says everyone was put on the waiting list, and then in the next paragraph it says half were assigned to Intervention group and half to the Waiting List Control group. Please clarify this and parse it down for clarity.
- I am confused since it says in lines 137-138 that 281 subjects were put on the waiting list control group for the study, but ultimately there were only 30 participants. Can you better clarify how this subgroup was selected for the intervention?
- Stage and phase are used interchangeably. Please choose only one and correct throughout the rest of the manuscript.
- When authors say that the data was collected at the beginning of the study in lines 159-161, is this referring to the beginning of the first phase of the study or was data collected again for these individuals? Which data was collected at which time points?
- Intervention Group and Experimental Group are both used. Please choose one and correct throughout the manuscript.
- There seems to be an inherent flaw with the hair cortisol data in that it measures 3 months of stress, and it seems the study only lasted for 9 weeks, which means some of the data at pre- and post-intervention (~1 month) would be overlapping. This needs to be mentioned as a serious Limitation in the physiological data collection here.
- It would be more helpful to describe who the experienced facilitators were (psychologists, spiritual teachers, etc?) in Section 2.7. Also, please clarify when the 4-hour immersion occurred.
- The language “guaranteeing a state of mindfulness” seems to be too strong a claim. It is not clear that something as subjective as a mindful state could be “guaranteed” from an outside observer, so I think this language needs to be adjusted (e.g., increasing the likelihood of participants achieving a mindful state during sessions).
- How often were the participants instructed to practice in between sessions?
- “Therefore,” can be removed at the beginning of section 2.8.
Results/Discussion/Conclusion
- Is there any way to include the data or some analyses related to how much the amount of between session practice impacted the results? If you have the data, it would be very meaningful.
- The statement made in lines 310-312 in relation to evidence-based treatments makes it sound as though this study was compared to another treatment. Please alter the language here or remove this sentence to avoid confusion.
- The finding that the control group cortisol increased is a very important contribution to the literature, and there can be some discussion added regarding the ways that mindfulness can be preventative for this group.
- The attempted explanation for findings that changes to physiological data was greater than that of the changes to the psychometric findings (line 327-333) was somewhat confusing (i.e., that physiological data precedes the psychometric data), particularly since the PSS and BAI seem to be more acute measures of psychological distress while hair cortisol measures more chronic stress, so these might actually be reversed. I might offer that there is some literature suggesting that mindfulness can actually increase one’s awareness of subjective anxiety and stress more acutely since they are no longer avoiding it, which makes one better able to cope with it long-term but might mask some of the findings of self-reported data of these constructs. Some of the studies below might provide some guidance on developing that argument:
Abramowitz, J. S., Deacon, B. J., & Whiteside, S. P. (2019). Exposure therapy for
anxiety: Principles and practice. Guilford Publications.
Hayes, S. C., Wilson, K. G., Gifford, E. V., Follette, V. M., & Strosahl, K. (1996).
Experiential avoidance and behavioral disorders: A functional dimensional
approach to diagnosis and treatment. Journal of Consulting and Clinical Psychology,
64(6), 1152.
- A statement is made in the Conclusion related to including mindfulness practice in the University Workers’ routines, but there is no data to suggest that mindfulness occurred more than once a week. I think that data needs to be included or that statement needs to be altered.
- In addition to including some of the limitations discussed above, please also move this from the Conclusion to the Discussion.
Author Response

(The authors gave the same response as above.)

Reviewer 3 Report
Comments and Suggestions for Authors
Dear Authors,
I read the article entitled “Mindfulness Practice Reduces Hair Cortisol, Anxiety and Perceived Stress in University Workers: a Randomized Clinical Trial”.
First of all the manuscript is clear, scientifically correct and presented in a well-structured manner.
The results and tables are clear, easy to interpret and understand, and the conclusions are consistent with the evidence and arguments presented.
Here are my observations:
1. The Abstract should indicate the sample size.
2. Bibliographic references both in the text and in the References section should be in accordance with the guidelines for authors suggested by the journal, i.e:
- References: References must be numbered in order of appearance in the text (including table captions and figure legends) and listed individually at the end of the manuscript. We recommend preparing the references with a bibliography software package, such as EndNote, ReferenceManager or Zotero to avoid typing mistakes and duplicated references. We encourage citations to data, computer code and other citable research material. If available online, you may use reference style 9. below.
- Citations and References in Supplementary files are permitted provided that they also appear in the main text and in the reference list.
In the text, reference numbers should be placed in square brackets [ ], and placed before the punctuation; for example [1], [1–3] or [1,3]. For embedded citations in the text with pagination, use both parentheses and brackets to indicate the reference number and page numbers; for example [5] (p. 10). or [6] (pp. 101–105).
Good luck
Author Response
Dear Reviewer
We appreciate your comments to increase the quality of our manuscript. Please find attached the responses to your suggestions.
Kind regards,
Authors

Round 2
Reviewer 1 Report
Comments and Suggestions for Authors
The authors hardly responded to my comments because they made almost no substantive modifications.
Author Response
Dear Reviewer
Thank you for you comments. We have adjusted the paper according to your recommendations.
Authors
